# α-Tocopherol Pharmacokinetics in Adults with Cystic Fibrosis: Benefits of Supplemental Vitamin C Administration

**DOI:** 10.3390/nu14183717

**Published:** 2022-09-09

**Authors:** Maret G. Traber, Scott W. Leonard, Vihas T. Vasu, Brian M. Morrissey, Huangshu (John) Lei, Jeffrey Atkinson, Carroll E. Cross

**Affiliations:** 1Linus Pauling Institute, Oregon State University, Corvallis, OR 97331, USA; 2Department of Zoology, The Maharaja Sayajirao University of Baroda, Vadodara 390002, Gujarat, India; 3Adult Cystic Fibrosis Program, Department of Internal Medicine, University of California, Davis, CA 95817, USA; 4Department of Chemistry, Center for Biotechnology, Brock University, St. Catharines, ON L2S 3A1, Canada; 5Department of Physiology and Membrane Biology, University of California, Davis, CA 95616, USA

**Keywords:** vitamin E, carboxyethyl hydroxy chromanol (CEHC), stable isotope-labeled vitamin E

## Abstract

Background: Numerous abnormalities in cystic fibrosis (CF) could influence tocopherol absorption, transportation, storage, metabolism and excretion. We hypothesized that the oxidative distress due to inflammation in CF increases vitamin E utilization, which could be positively influenced by supplemental vitamin C administration. Methods: Immediately before and after receiving vitamin C (500 mg) twice daily for 3.5 weeks, adult CF patients (*n* = 6) with moderately advanced respiratory tract (RT) disease consumed a standardized breakfast with 30% fat and a capsule containing 50 mg each hexadeuterium (d_6_)-α- and dideuterium (d_2_)-γ-tocopheryl acetates. Blood samples were taken frequently up to 72 h; plasma tocopherol pharmacokinetics were determined. During both trials, d_6_-α- and d_2_-γ-tocopherols were similarly absorbed and reached similar maximal plasma concentrations ~18–20 h. As predicted, during vitamin C supplementation, the rates of plasma d_6_-α-tocopherol decline were significantly slower. Conclusions: The vitamin C-induced decrease in the plasma disappearance rate of α-tocopherol suggests that vitamin C recycled α-tocopherol, thereby augmenting its concentrations. We conclude that some attention should be paid to plasma ascorbic acid concentrations in CF patients, particularly to those individuals with more advanced RT inflammatory disease and including those with severe exacerbations.

## 1. Introduction

Cystic Fibrosis (CF) is a genetic disorder, affecting approximately 40,000 people in the US and more than 160,000 world-wide [1,2]. Mutations in the Cystic Fibrosis Conductance Regulator (CFTR) gene give rise to decreased functional CFTR in epithelial cells, leading to desiccated and hyperviscous respiratory tract (RT) and gastrointestinal tract secretions [3]. Hallmarks of the disease include compromised RT mucociliary clearance, microbial infection and an exaggerated inflammatory response mainly by a florid RT neutrophilia and activations of neutrophil proteolytic and oxidative processes [4,5,6]. These latter processes are believed to play a central role in the progressive lung tissue destruction seen in CF, as indicated by strong evidence of active proteolytic [6,7,8] and oxidative [9,10,11,12,13,14,15,16,17,18,19] processes seen in RT of CF patients.

Nutritional deficiencies, in large part secondary to abnormalities of bile acid homeostasis and exocrine pancreatic insufficiency [20,21], represent important contributors to the CF clinical manifestations. Although current CF managements (replacement pancreatic lipase and lipophilic vitamin supplements) address the resulting severe lipid malabsorption and fat-soluble vitamin deficiencies associated with pancreatic insufficiency, these are only partially effective. There remain uncertainties as to the magnitudes of the absorption and plasma kinetics of lipophilic nutrients. Vitamin E is particularly relevant in CF because α- and γ-tocopherols represent major dietary chain breaking, lipophilic antioxidants. α-Tocopherol deficiency, in particular, is known to have deleterious clinical consequences in CF [22,23,24].

Many in vitro experimental models have convincingly documented interactions between the important lipophilic antioxidant, vitamin E, and the major hydrophilic antioxidant, vitamin C (ascorbic acid) [25,26,27,28,29]. The primary mechanism for this interaction is that the lower reduction potential of vitamin C is capable of reducing the accessible oxidative free radical of vitamin E (tocopheroxyl radical) in membranes [30], thus regenerating vitamin E [27]. This “recycling” phenomenon becomes of clinical relevance under circumstances of severe inflammatory processes known to generate reactive oxidants. This phenomenon could compromise ascorbic acid concentrations [31,32] and impact the capability to regenerate the tocopheroxyl radical. As ascorbic acid concentrations relate inversely to biomarkers of inflammation in CF [33], interplays of vitamin C and vitamin E could be of potential importance in this disease, particularly during periods of acute, infective exacerbations.

Our group has previously studied tocopherol pharmacokinetics in smokers, another group of subjects under considerable RT oxidative stress from inhalation of free radicals contained in cigarette smoke (CS) and subsequent RT inflammation-related oxidative injury [34,35,36,37]. Using deuterium-labeled α-tocopherol biokinetics similar to those used in the present studies, we showed that smokers compared to non-smokers exhibit a significant 25% decrease in the plasma α-tocopherol half-life (t ½) [37]. We further showed in another group of smokers and non-smokers that the smokers t ½ clearance rates for α-tocopherol were 40% faster than non-smokers, that the accelerated vitamin E plasma disappearance rates in smokers were inversely correlated with their ascorbic acid concentrations, and that the administration of a vitamin C supplement could decrease smokers’ accelerated plasma disappearance rates so as to be not significantly different that those of non-smokers while also decreasing their markers of lipid peroxidation [35,36,37].

Considering this background, it seemed appropriate to further quantitate the pharmacokinetics and metabolism of α- and γ-tocopherols in a small, representative group of adults with CF and considerable inflammatory RT disease, using a repeated measures design where participants serve as their own controls. Our focus is on the interplay of vitamin C supplementation on α- and γ-tocopherols plasma pharmacokinetics.

## 2. Materials and Methods

### 2.1. Participant Characteristics

Participants with CF (*n* = 6, ages 23–31 years y, 3 females and 3 males) were recruited from the UC Davis Adult CF Clinic. All subjects met diagnostic criteria for CF. No other specific inclusion or exclusion criteria were specified. All participants were deemed to be compliant with their standard outpatient CF regimens and all appeared to be clinically stable. The study protocol was approved by the institutional review boards for the protection of human subjects at UC Davis (#200715896-3) and at Oregon State University (OSU IRB protocol #3988). All participants gave written, informed consent. Studies were conducted prior to patients receiving either CFTR potentiator or corrector therapy, as described below. The clinical trial was carried out in 2008–2009 prior to requirements for listing in Clinical Trials.gov.

### 2.2. Deuterium-Labeled Vitamin E

α-5,7-(CD_3_)_2_ tocopheryl acetate (d_6_-α-tocopheryl acetate) and unlabeled α-tocopheryl acetate (d_0_-α-tocopheryl acetate) were kind gifts from Dr. James Clark of Cognis Nutrition and Health. γ-3,4-(D) tocopheryl acetate (d_2_-γ-tocopheryl acetate) was prepared from γ-tocopherol labeled with two deuterium atoms, as described [38]. The d_6_-α- to d_2_-γ-tocopherol molar ratio was determined by liquid chromatography/mass spectrometry (LC/MS) to be 0.98. The internal standard used for mass spectroscopy, α-5,7,8-(CD_3_)_3_ tocopheryl acetate (d_9_-α-tocopheryl acetate), was provided by Dr. Carolyn Good of The Bell Institute of Health and Nutrition, and was synthesized by Isotec, Inc. (Miamisburg, OH, USA).

### 2.3. Study Design and Blood Sampling

In brief, the protocol included a preliminary screening visit with an initial plasma inflammatory “profile” and clinical exam. The protocol was explained in detail to the six volunteers, who gave written informed consent at the UC Davis Translational Research Clinical Center at the Mather VA Hospital, Sacramento, CA, USA. A blood sample was obtained, and the participants were instructed to maintain their ordinary and usual lifestyle, CF management regimens, habitual diets, including their pancreatic-lipase, and A, D, E, and K vitamin preparations (no record of specific formulations is available). They were asked to not partake of any additional non-conventional complimentary nutritional supplements beyond their prescribed high-calorie nutritional supplements. General demographic data, pulmonary function, and respiratory tract cultures of the participants were obtained within 90 days of the 3 study visits and routine clinic hospital clinical laboratory blood measurements (Table 1). The clinical laboratory measured vitamin A as retinol and vitamin D as 25-hydroxyvitamin D by standard protocols.

Two weeks after the preliminary screening visit, the subjects returned to the Research Clinical Center and were provided with a standardized breakfast (600 Kcal with 30% of the calories from fat), took their standard pancrealipase preparation followed by oral administration of a capsule containing an equal molar mixture of approximately 50 mg each of d_6_-α- and d_2_-γ-tocopherol acetates. Blood samples (*n* = 9 per trial) were collected prior to breakfast and at 3, 6, 9, 12, 24, 36, 48 and 72 h after taking the vitamin E capsule. At the 72 h visit to the Clinical Center the subjects were given vitamin C (500 mg tablets) to be taken twice daily for 3 ½ weeks, at which time they returned to the Clinical Center to repeat the 72 h vitamin E pharmacokinetic study.

Blood samples were collected from the antecubital vein into evacuated tubes containing either 0.05 mL 15% (wt:vol) EDTA or sodium heparin (Becton Dickinson, Franklin Lakes, NJ, USA). Plasma was promptly separated by centrifugation at 4 °C for 15 min at 500× *g* and stored at −80 °C until analyzed. After heparinized plasma was separated, an aliquot was acidified (1:1) with 10% PCA (perchloric acid) containing 1 mM diethylenetriaminepentaacetic acid (DTPA). This sample was then centrifuged (5 min, 15,000× *g*, 4 °C) the supernatant removed, frozen in liquid nitrogen, and stored at −80 °C until analysis.

### 2.4. Laboratory Analyses

Labeled and unlabeled plasma α- and γ-tocopherols were extracted [39] and measured by liquid chromatography/mass spectrometry (LC/MS) using negative atmospheric pressure chemical ionization (-APCI) as previously described [34,40]. Plasma ascorbic acid, following plasma acidification, was measured by HPLC with amperometric detection as previously described [26]. Plasma triglycerides and total cholesterol were determined by standard clinical assays (Sigma, St. Louis, MO, USA). Plasma vitamin E catabolites (CEHCs) were extracted using a modified method [41] and analyzed by LC/MS using negative electrospray ionization (-ESI) as previously described [34]. For use as standards 2,5,7,8-tetramethyl-2-(2′-carboxyethyl)-6-hydroxychroman (α-CEHC) and 2,7,8-trimethyl-2-(β-carboxyethyl)-6-hydroxychroman (γ-CEHC) were obtained (Sigma-Aldrich, St Louis, MO, USA).

Plasma malondialdehyde (MDA) concentrations were measured as described [42]. Quantitation was done using an external standard of 1,1,3,3-tetraethoxypropane (Sigma) prepared using the same method. The MDA-TBA adduct was extracted with butanol and measured by HPLC with fluorometric detection (532 nm excitation and 553 nm excitation). The TBA-MDA adduct was quantified against the MDA standards.

### 2.5. Mathematical and Statistical Analyeis

The maximum tocopherol and CEHC concentrations (Cmax) and the time of maximum concentration (Tmax) were determined by visual inspection of each participant’s data. Plasma exponential disappearance rates of d_6_-α-tocopherol, d_2_-γ-tocopherol and d_2_-γ-CEHC were estimated as previously described [34]. Half-lives of these compounds were calculated as t ½ = ln (2)/exponential disappearance rate constant. One-sided, paired t-tests were used to compare values from the baseline trial to the vitamin C supplemented trial (Excel, Microsoft). Two-factor analysis of variance with repeated measures was used to assess changes over time within subjects (Prism 6 for Macintosh, GraphPad). Data are shown as means ± standard deviation (SD), *n* = 6, unless as otherwise noted.

## 3. Results

### 3.1. Baseline CF Subject Characteristics

Demographic profiles of the six CF participants are as depicted in Table 1. Note that four persons were homozygous for the most frequent 508/508 genotype [43]. All had moderate to severe lung function abnormalities based on forced expiratory volume and all had suboptimal nutritional status as measured by BMIs. Although one subject was a diabetic requiring insulin administration, all had normal HbA1c values. All subjects had pancreatic insufficiency, and all were taking conventional doses of pancreatic enzymes and a standard ADEK lipophilic micronutrient supplement. Vitamins A and D micronutrient concentrations, as determined in the clinical laboratories within several months of the study protocol, were generally in the low normal range, whereas inflammatory parameters, as reflected by the latest C-reactive protein (CRP) in juxtaposition to the study itself, showed a significant degree of variation. Table 2 depicts the average values of each of the subjects’ total cholesterol, triglycerides and lipid (sum of cholesterol and triglycerides) concentrations during the trials, as well as plasma α-tocopherol concentrations prior to each of the two kinetic studies. Also shown are the calculated ratios of α-tocopherol per total cholesterol and α-tocopherol per total lipids.

### 3.2. Efficacy of Vitamin C Supplementation

The baseline pharmacokinetic trial (control) was carried out, followed by 3.5 weeks of vitamin C supplementation, then the pharmacokinetic trial was repeated. Vitamin C supplements were effective in increasing plasma ascorbic acid concentrations (*p* = 0.0023, Figure 1a). MDA was measured to assess oxidative stress; these values were not significantly changed between the two trials (Figure 1b), although they were somewhat lover after vitamin C supplementation.

### 3.3. Vitamin E Pharmacokinetics

Plasma α- and γ-tocopherol concentrations measured during the trials were low (Table 3) and on average are in the deficient range [44]. Only one participant (#3) had values of ~20 µmol/L; the others ranged from 5 to 16 µmol/L, suggesting that the vitamin E supplementation was inadequate.

Participants consumed the deuterated tocopherols with a standard breakfast, then blood samples were taken up to 72 h. A representative subject’s data illustrates that both deuterated α- and γ-tocopherols were similarly absorbed and appeared in the plasma with similar maximum concentrations (Figure 2). Vitamin C supplementation had no impact on the time of maximum concentrations (Tmax and Cmax) for either of the tocopherols (Table 3). Importantly, however, the exponential disappearance rates of α-tocopherol disappearance from the plasma were significantly slower (*p* < 0.05) during the vitamin C supplementation (Table 3).

The half-lives, calculated from the exponential disappearance rates, show that α-tocopherol has a half-life that is nearly double that of γ-tocopherol (Figure 3). Vitamin C supplementation prolonged the retention of α-tocopherol in the plasma, but not that of γ-tocopherol. Vitamin C supplementation had no effect on the rates of disappearance of either γ-tocopherol or its catabolite, γ-CEHC. It should be noted that not only were γ-tocopherol rates of disappearance rapid, they were also similar to those of γ-CEHC, suggesting the importance of vitamin E catabolism of γ-tocopherol, even in vitamin E deficient persons.

The entire data set were also calculated based on percentage labeled of either the total plasma α- or of γ-tocopherol concentrations, respectively. The purpose was to verify that the low plasma α-tocopherol concentrations observed in some participants did not affect the exponential rates of disappearance. The individual responses shown in Figure 3b,c illustrate the changes in response to vitamin C supplementation. Specifically, the exponential disappearance rates for d_6_-α-tocopherol were slower, but those for d_2_-γ-tocopherol were unchanged by vitamin C supplementation. These outcomes are similar to the responses to vitamin C supplementation that were observed when the rates were calculated based on plasma d_6_-α- and d_2_-γ-tocopherol concentrations.

## 4. Discussion

Vitamin C and E, representing the major dietary hydrophilic and lipophilic antioxidant micronutrients, respectively, have long been known to interact as elegantly demonstrated in chemical in vitro studies [27] and in selected clinical studies [35,37]. The current study demonstrates that, similar to our previous findings in cigarette smokers [35,37], this interrelationship can be demonstrated in moderately severe stable chronic suppurative CF RT disease patients. Importantly, the plasma concentrations observed in CF in response to a 50 mg dose of each d_6_-α- and d_2_-γ-tocopheryl acetates were much lower than observed in healthy subjects in other studies. For example, Bruno et al. reported that healthy adults with plasma α-tocopherol of approximately 20 μmol/L, who consumed 22 mg d_6_-α-tocopheryl acetate with increasing levels of fat, showed a peak plasma d_6_-α-tocopherol concentration of 5.6 ± 1.2 μmol/L (Table 2, ref. [45]). The participants with CF reported herein (Table 3) received a 50 mg dose of d_6_-α-tocopheryl acetate, but their maximum d_6_-α-tocopherol concentration was 10-times lower—approximately 0.5 μmol/L. Leonard et al. also tested a 50 mg dose of each d_6_-α-tocopheryl acetate and d_2_-γ-tocopherol in healthy adults. They reported that plasma d_2_-γ-tocopherol peaked at lower concentrations (*p* < 0.001; 2.2 ± 1.2 μmol/L than did d_6_-α-tocopherol (6.0 ± 2.6 μmol/L) [34]. Again, these concentrations are 10-times higher than those reported herein, suggesting that vitamin E absorption in participants with CF was impaired. Notably, the average plasma α-tocopherol concentrations in the present study (Table 2) were at or near the deficient level, even when corrected for circulating lipid levels [44].

Vitamin E deficiency in humans can be devastating due to its neurologic consequences. Thus, supplemental vitamin E doses are larger than dietary recommendations for normal healthy persons [44]. Vitamin E deficiency symptoms were first described in children with fat malabsorption syndromes, principally abetalipoproteinemia, cystic fibrosis and cholestatic liver disease [46]. Subsequently, humans with severe vitamin E deficiency with no known defect in lipid or lipoprotein metabolism were found to have a defect in the gene for the α-tocopherol transfer protein (α-TTP). This syndrome is called “Ataxia with Vitamin E Deficiency” or AVED. The neurologic abnormalities due to vitamin E deficiency in AVED are described as a progressive sensory neuropathy that can be halted and in some cases reversed by vitamin E supplements. Enormous daily supplemental α-tocopherol amounts (>100 mg/kg body weight) given long-term can overcome the lack of apoB-lipoproteins in abetalipoproteinemia [47], to prevent neurologic disease progression [48] and to prevent oxidative damage [49]. Similarly, supplemental α-tocopherol (1000 mg/day) can prevent progression of neurologic defects in AVED [50]; one patient has been reported to be stable for over 30 years [51]. Persons with fat malabsorption due to impaired biliary secretion generally do not absorb orally administered vitamin E. They are treated with special forms of vitamin E, such as α-tocopheryl polyethylene glycol succinate, which spontaneously form micelles, obviating the need for bile acids [52]. Previous studies have suggested that vitamin E supplements needed to be 400 mg daily in order to raise CF plasma concentrations in adults to approximately 20 μmol/L [53].

Only one of the CF participants studied herein had plasma α-tocopherol concentrations at this normal level. Winklhofer-Roob et al. showed in children (average age 9.2 years) with CF that after an overnight fast, 100 mg α-tocopheryl acetate given with whole milk and an optimized dose of pancreatic enzymes (sufficient to correct fat maldigestion), showed a plasma α-tocopherol increase of nearly 10 μmol/L [54]. Despite the apparent low absorption in the present study, the fractional disappearance rates (0.5 pools/day) reported herein for CF post-vitamin C supplementation (Table 3) are similar to those reported for healthy participants (0.49–0.56 pools per day [45]). Previously, doses as low as 1 to 2 mg have been used successfully for measuring vitamin E kinetics [55,56]. Thus, the absorbed dose size, based on previous studies, does not impact disappearance rates.

As shown in Figure 2 and Table 3, we did indeed demonstrate significant synergism between α-tocopherol and vitamin C. It is particularly important to note that the stable CF patient group, prior to their vitamin C supplementation, had adequate plasma ascorbic acid concentrations (~60 μmol/L), which increased with supplementation to ~80 μmol/L. Moreover, none of the participants experienced clinical exacerbations during the study.

### 4.1. Nuances for the CF Community

The CF care community has long been aggressively focused on the needs for antioxidant micronutrient supplements [12,13,14,16,17,57,58,59], and particularly for the needs for lipophilic antioxidants such as vitamin E [17,18,23,24,60,61,62,63]. Under strong inputs from the CF Foundation scientific community [14,16], a revised version of a micronutrient antioxidant cocktail was designed and is now marketed with this increased need in mind [58]. Although studies have shown CF patients [33]; like smokers [64], have low or low normal levels of vitamin C, no augmentations of vitamin C were included in the new formulation. The recent report of significant improvements in the endothelial dysfunction of CF patients, as reflected by increases in measured brachial artery flow mediated dilatation after ingestion of an antioxidant cocktail containing 1000 mg of vitamin C, suggests that present supplemented antioxidants may not be optimized [65].

Although it is known that CF exacerbations are associated with increases in their baseline levels of oxidative stress [66,67] accompanied by further decreases in their levels of antioxidant micronutrients, including vitamin E [33,66,68], the existing CF exacerbation treatment guidelines do not address needs for increased antioxidant supplements during exacerbations [58,69,70,71].

Of note, the CF community has already recognized the value of standard of care monitored measurements of the lipophilic vitamins A, D, and E and probably would measure vitamin C, if easily determinable in a clinic or hospital setting. The emerging data from both smokers and critically ill patients [72,73,74,75] suggest the probable value of including vitamin C measurements and augmentations in CF patients undergoing hospital care for severe exacerbations due to their acute effects on chronic airway septic exacerbations. These severe CF exacerbations are frequently accompanied by nutritional deficiencies. To date, such measurements are usually not done because of the technological challenges of accurate ascorbic acid determinations in clinical laboratories which have not been widely used to determine ascorbic acid concentrations [75].

A prospective study to determine whether vitamin C supplementation in CF would be beneficial, particularly in those CF patients with advanced and/or exacerbated RT disease, should be considered, much as has been proposed for anti-inflammatory agents [76] and done for the case of doxycycline [77].

### 4.2. Limitations

Several limitations to the scope of the present study in a small number of CF participants should be mentioned. Most CF patients are now on revolutionary CFTR potentiator and/or corrector therapies and these are likely to exert impacts on vitamin E pharmacokinetic via potential effects on bile and pancreatic secretions, enterocyte functions and modulations of the intensity of RT inflammatory processes [78,79,80,81,82,83,84,85]. No attempts were made in the present study to relate plasma tocopherol kinetics to either parameters of RT or systemic inflammation/oxidative stress or to acute or chronic CF exacerbations. Finally, it should be recognized that interrelationships between vitamins E and C, as revealed by kinetics in the plasma compartment, are not likely to reflect the degrees of their interdependence in other extracellular sites. For example, considerable evidence supports the concept that oxidative processes are taking place in the RT of CF patients [10,16]. The goal of antioxidant micronutrient therapy in severe CF RT disease would seem more likely to restore proper levels in RT itself rather than plasma ascorbic acid concentrations. Caution needs to be exercised in extrapolations of their interdependence from blood to such complex redox milieu as exists in the inflamed CF RT.

## 5. Conclusions

Our major finding was that vitamin C potentiates the biological availability of vitamin E in the plasma compartment of CF patients with advanced but stable RT inflammatory disease. This outcome supports previous observations made in cigarette smokers who are known to have CS-related RT inflammatory changes. It would seem to be clinically prudent for CF clinicians and nutritionists to recognize the importance of dietary and supplemental vitamin C levels and their possible impact on plasma α-tocopherol kinetics in their patients who are undernourished and/or experiencing severe exacerbations of their CF RT inflammatory disease.

## Figures and Tables

**Figure 1 nutrients-14-03717-f001:**
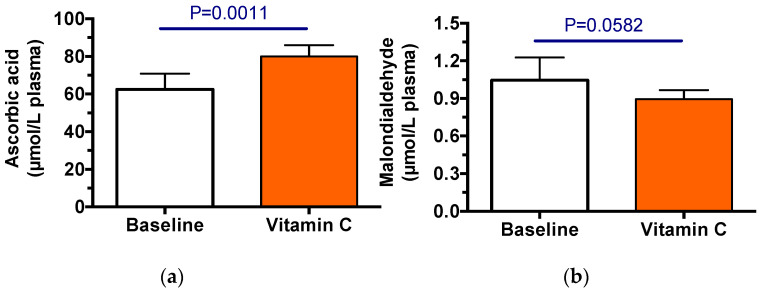
The efficacy of vitamin C supplementation in CF participants. Plasma concentrations of ascorbic acid (**a**) and malondialdehyde (**b**) during pharmacokinetic trials. Shown are the means (±SD, *n* = 6) of the average concentrations measured from plasma samples collected at 0, 3, 6, 9, 12, 24, 48, 72 h from each participant.

**Figure 2 nutrients-14-03717-f002:**
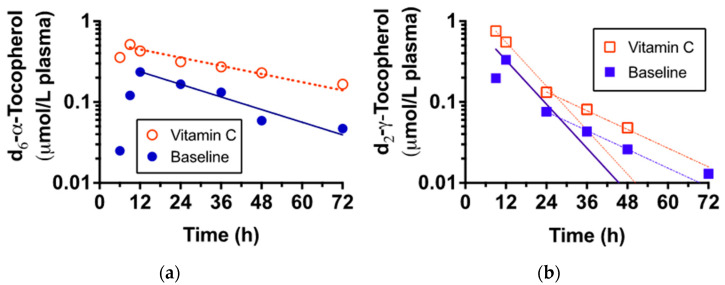
The plasma d_6_-α- (**a**) and d_2_-γ-tocopherol (**b**) concentrations at baseline and after vitamin C supplementation (representative participant, #3). Plasma-labeled and unlabeled tocopherols were measured by LC/MS from blood samples periodically collected up to 72 h. Filled symbols denote baseline, open symbols denote vitamin C pharmacokinetics trial. Lines indicate post-peak exponential decay curves. The d_2_-γ-tocopherol rates of disappearance were so fast that the slopes were no longer linear after 36 h; thus, a second curve was fit to the data. Neither slope was altered by vitamin C status; only the slope from Tmax is reported.

**Figure 3 nutrients-14-03717-f003:**
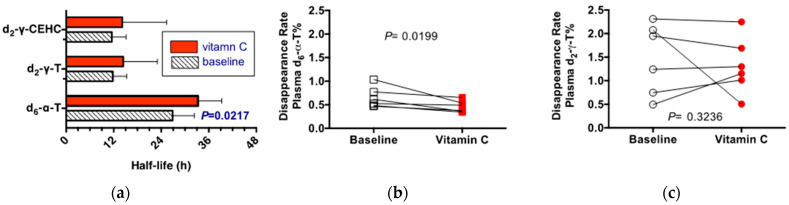
The plasma half-lives in hours for d_6_-α- and d_2_-γ tocopherols and d_2_-γ CEHC. Half-lives indicate the length of time for half of the indicated compound to leave the plasma compartment (**a**). Individual data shows disappearance rates (pools per day) by person (**b**) d_6_-α tocopherol; (**c**) d_2_-γ tocopherols). The rates shown are based on the percentage labeled to demonstrate that the baseline tocopherol concentrations did not impact the outcomes.

**Table 1 nutrients-14-03717-t001:** CF participant demographics.

#	Age (Years)	Sex	CF Genotype	BMI	Predicted FEV1 ^1^	CRP ^2^ (mg/dL)	WBC	Hb	HbA1C	Vit. A ^3^ (mg/L)	Vit. D ^4^ (ng/mL)	Sputum ^5^
1	28	M	508/508	20	38	---	5000	14.2	5.1	0.37	20	*Pseudomonas*
2	32	F	508/508	20	34	2.0	8700	9.8	5.3	0.40	25	*Pseudomonas*
3	31	M	508/1717-G→A	22	66	0.2	8000	12.6	5.4	0.72	24	*Pseudomonas*/*S. aureus*
4	30	F	508/508	21	46	1.3	8000	12.7	6.5	0.34	29	*Pseudomonas*/MRSA
5	25	F	508/508	15	30	6.7	7700	12.8	6.6	0.20	34	*S. maltophilia*
6	23	M	508/711+1 G→T	20	47	0.6	15,600	12.4	6.1	0.30	49	*Pseudomonas*/MRSA

^1^ forced expiratory volume in 1 s (FEV1), ^2^ C-reactive protein (CRP), ^3^ Vitamin A, ^4^ Vitamin D, ^5^
*Staphylococcus aureus (S. aureus)*, methicillin-resistant *S. aureus* (MRSA), *Stenotrophomonas maltophilia (S. maltophilia*).

**Table 2 nutrients-14-03717-t002:** The participants’ lipid concentrations during baseline and vitamin C interventions.

	Triglycerides (mmol/L)		Cholesterol (mmol/L)		Total Lipids (mmol/L)		α-Tocopherol (µmol/L)		α-T/Cholesterol (mmol/mol)		α-T/Lipids (mmol/mol)	
#	Baseline	Vitamin C	Baseline	Vitamin C	Baseline	Vitamin C	Baseline	Vitamin C	Baseline	Vitamin C	Baseline	Vitamin C
1	0.48	0.42	2.61	2.39	3.09	2.82	6.99	5.03	2.66	2.11	2.25	1.80
2	0.41	0.41	3.62	3.30	4.03	3.72	17.91	16.17	4.96	4.89	4.45	4.35
3	0.93	0.87	3.86	3.40	4.79	4.27	20.73	19.08	5.39	5.61	4.34	4.47
4	0.32	0.34	4.34	3.78	4.65	4.13	13.41	15.16	3.09	4.03	2.88	3.69
5	0.28	0.27	2.21	2.15	2.49	2.42	7.18	9.51	3.26	4.42	2.89	3.93
6	0.35	0.28	2.50	2.68	2.86	2.95	8.61	7.72	3.47	2.89	3.05	2.62

Cholesterol and triglyceride concentrations did not differ significantly between the two trials; however, their sum shown as total lipids (3.65 ± 0.97 vs. 3.38 ± 0.76) was slightly lower during the vitamin C (Vit C) intervention (*p* = 0.0467). Plasma unlabeled α-Tocopherol (α-T), α-T/cholesterol and α-T/lipids were not significantly different between the two interventions. Lipid levels were not used to modify vitamin E pharmacokinetics.

**Table 3 nutrients-14-03717-t003:** The vitamin E concentrations and pharmacokinetic parameters.

		Intervention		
Plasma		Baseline	Vitamin C	Paired *t*-Test
α-Tocopherol	Average concentration (µmol/L plasma)	12.5 ± 5.9	12.1 ± 5.5	NS
γ-Tocopherol		0.52 ± 0.38	0.60 ± 0.51	NS
α-CEHC *	Average concentration (µmol/L plasma)	0.12 ± 0.09	0.19 ±0.08	NS
γ-CEHC		0.47 ± 0.53	0.59 ± 0.80	NS
d_6_-α-Tocopherol ^1^	Cmax (µmol/L)	0.27 ± 0.15	0.32 ± 0.13	NS
	Tmax (h)	19.5 ± 7.0	17.5 ± 7.5	NS
	Disappearance rate (pools per day)	0.65 ± 0.14	0.50 ± 0.10	*p* = 0.0263
	AUC	9.78 ± 5.57	12.04 ± 4.38	NS
d_2_-γ-Tocopherol	Cmax (µmol/L)	0.20 ± 0.10	0.30 ± 0.22	NS
	Tmax (h)	15.5 ± 6.7	11.5 ± 1.2	NS
	Disappearance rate (pools per day)	1.54 ± 0.50	1.42 ± 0.55	NS
	AUC	4.59 ± 3.14	5.66 ± 2.77	NS
d_2_-γ-CEHC *n* = 5	Cmax (µmol/L)	0.20 ± 0.14	0.22 ± 0.10	NS
	Tmax (h)	16.2 ±7.2	12.0 ± 0	NS
	Disappearance rate (pools per day)	1.25 ± 0.72	1.61 ± 0.79	NS

The data shown is calculated from *n* = 6 participants. Plasma labeled and unlabeled α- and γ-tocopherols were measured simultaneously in the same sample. Plasma labeled and unlabeled α- and γ-CEHCs were measured simultaneously in a separate analysis from the tocopherols. * Unlabeled γ-CEHC was detectable in all subjects’ samples. No d_6_-α-CEHC was detected in any plasma sample and several samples had no detectable unlabeled α-CEHC. During the baseline trial, α-CEHC was undetectable at multiple times in subjects 1 (once), 5 (at all times) and 6 (twice); while during the vitamin C trial α-CEHC was undetectable in all plasma samples in subjects 1, 5, and 6. During the baseline trial, subject 1 had no detectable d_2_-γ-CEHC, so this subject’s data is not included for d_2_-γ-CEHC averages shown in the table (*n* = 5). ^1^ The d_2_-γ- and d_6_-α-tocopherol disappearance rates were different (*p* < 0.001) from each other in each trial, while the Cmax were not different, indicating similar absorption, but faster disposition of d_2_-γ-tocopherol. The fast disposition was also supported by the d_2_-γ-CEHC disappearance rates, which were similar to those of d_2_-γ-tocopherol.

## Data Availability

Not applicable.

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
