# Peer review of "α-Tocopherol Pharmacokinetics in Adults with Cystic Fibrosis: Benefits of Supplemental Vitamin C Administration"

_nutrients, 2022, doi:10.3390/nu14183717_

Round 1

Reviewer 1 Report

Very interesting and important paper. There is no similar paper in the scientific literature. English language is correct. Investigated processes play major role in the progressive lung tissue destruction. The vitamin C-induced decrease in the plasma disappearance rate of alpha-tocopherol suggests that vitamin C re-cycled alpha-tocopherol, thereby augmenting its concentrations. The Authors concluded that some attention should be paid to plasma ascorbic acid concentrations in CF patients, particularly to those individuals with more advanced RT inflammatory disease including those with severe exacerbations. You could add to the introduction the paper: Hamilton IM, Gilmore WS, Benzie IF, Mulholland CW, Strain JJ. Interactions between vitamins C and E in human subjects. Br J Nutr. 2000 Sep;84(3):261-7. doi: 10.1017/s0007114500001537. PMID: 10967604. This study was designed to examine the effects of supplementation with either vitamin C or E on their respective plasma concentrations, other antioxidants, lipids and some haemostatic variables.

Author Response

Authors’ Response: Thank you for your kind comments and evaluation of our manuscript. We respectfully disagree that the cited paper is important to include. The paper by Hamilton et al shows that the E/lipids ratio improved post-vitamin C supplementation, but this was largely due to decreases in serum lipids (both cholesterol and triglycerides). Since these measures can be influenced by vitamin C (see: McRae MP. Vitamin C supplementation lowers serum low-density lipoprotein cholesterol and triglycerides: a meta-analysis of 13 randomized controlled trials. J Chiropr Med. 2008 Jun;7(2):48-58. doi: 10.1016/j.jcme.2008.01.002. PMID: 19674720; PMCID: PMC2682928.) It is unclear if these changes are due to methodologic problems (Nah H, Yim J, Lee SG, Lim JB, Kim JH. Ascorbate Oxidase Minimizes Interference by High-Concentration Ascorbic Acid in Total Cholesterol Assays. Ann Lab Med. 2016 Mar;36(2):188-90. doi: 10.3343/alm.2016.36.2.188. PMID: 26709272; PMCID: PMC4713858.) Therefore, we have chosen not to include the paper to avoid the controversy concerning lipids since these were not changed significantly in our study.

Reviewer 2 Report

The submitted manuscript by Traber et al is an interesting pharmacokinetic study in a rare disease cystic fibrosis. I think that this paper is sufficiently strong for publication in Nutrients, even if the number of patients is very low (6). There are some issues which should be resolved before possible acceptance of this paper:

Authors economized somehow with methodological section.

·         The method for analysis of vitamin A and vitamin D reported in table 1 is missing (e.g. what was precisely measured, I suppose retinol and hydroxycholecalciferol ?), add please a section reporting this

·         A similar lack is in the case of vitamin C? total vitamin C (including dehydroascorbic acid), ascorbic acid solely?

Results

There are some shortcomings with the kinetic study of radioactive vitamin E:

Table 3 – what does it mean rate? Why y-intercept is reported. This has little sense for a pharmacokinetic study. Report, please, AUC and elimination half-lives.

Figure 2 – this is strange. A pharmacokinetic graph with normal scale should show exponential decay in concentration, but in the case of alpha-tocopherol, there is linear relationship (this is normally observed in logarithmic scale). This is unclear. The graph for gamma-tocopherol is non correct. The curve should have fallen to 0 after 72 hours.  Add also error bars to the graph, they can be one-sided.

Figure 3 – B and C, clear and acknowledged pharmacokinetic parameters should be reported, remove b and c or clearly explain the meaning and importance for this study.

Small comments:

p. 226 -  Previous studies have suggested that vitamin E supplements needed to be 400 mg - This should be carefully discussed as doses 400 IU (400 mg of d/l tocopherol) can be associated with harmful effect - https://pubmed.ncbi.nlm.nih.gov/15537682/, discuss please.

Remove sentence „Our previous work also demonstrated that cigarette smokers exhibited an accelerated turnover of tocopherols in the plasma compartment and that vitamin C supplements restored the turnover rate in the plasma to that of non-smokers [34,35,37].“ from rows 238-240. This is repetition of the introduction and here it does not bring a clear contribution to the discussion.

Just a curiosity, the clinical study was performed 2008-2009, what is the reason for such a long distance between the study and this paper?

Author Response

Response to Reviewer 2:

The submitted manuscript by Traber et al is an interesting pharmacokinetic study in a rare

disease cystic fibrosis. I think that this paper is sufficiently strong for publication in Nutrients, even if the number of patients is very low (6).

Authors’ Response: We thank the reviewer for the assessment of our manuscript.

There are some issues which should be resolved before possible acceptance of this paper:Authors economized somehow with methodological section. The method for analysis of vitamin A and vitamin D reported in table 1 is missing (e.g. what was precisely measured, I suppose retinol and hydroxycholecalciferol ?), add please a section reporting this

Authors’ response: Lines 118-19 have been added: The hospital clinical laboratory measured vitamin A as retinol and vitamin D as 25-hydroxyvitamin D by standard protocols.

  • A similar lack is in the case of vitamin C? total vitamin C (including dehydroascorbic acid), ascorbic acid solely?

Authors’ response: Serum is acidified and therefore ascorbic acid is measured in our assay. This is now clarified in line 141.

Results

There are some shortcomings with the kinetic study of radioactive vitamin E:

Authors’ response: Please note this was not a study of radioactive isotopes; we used deuterium labeled vitamin E and measured the actual concentrations of labeled and unlabeled tocopherols.

Table 3 – what does it mean rate? Why y-intercept is reported. This has little sense for a pharmacokinetic study. Report, please, AUC and elimination half-lives.

Authors’ response: The AUC is now reported and the y-intercept deleted. The AUC is not significantly impacted by vitamin C. The Cmax and the time on the x-axis are the same for baseline and Vitamin C studies, only the slope changes, so little change is seen in AUC. The half-life is reported in Fig 3 to emphasize the comparisons between alpha and gamma tocopherols. The rate is the disappearance rate from the plasma, not necessarily the elimination from the body. Tocopherol kinetics are confounded by the recirculation of the alpha-tocopherol that is promoted by the alpha-tocopherol transfer protein. Please see our previous publications, especially: Traber MG, Ramakrishnan R, and Kayden HJ. Human plasma vitamin E kinetics demonstrate rapid recycling of plasma RRR-alpha-tocopherol. Proc Natl Acad Sci U S A. 1994;91(21):10005-8.

Figure 2 – this is strange. A pharmacokinetic graph with normal scale should show exponential decay in concentration, but in the case of alpha-tocopherol, there is linear relationship (this is normally observed in logarithmic scale). This is unclear. The graph for gamma-tocopherol is non correct. The curve should have fallen to 0 after 72 hours.  Add also error bars to the graph, they can be one-sided.

Authors’ response: Note Figure 2 is shown as a log-scale on the y-axis. Figure 2b is replaced with a graph showing that two different exponential lines fit the data; only the slope of the first is reported. It is impossible to fall to 0 on a log scale, so it is unclear what the reviewer meant. Also the error terms are reported for the comparisons between subjects; shown is a representative subject.

Figure 3 – B and C, clear and acknowledged pharmacokinetic parameters should be reported, remove b and c or clearly explain the meaning and importance for this study.

Authors’ response: The data in Fig 3b and c is to show individual responses. The section between lines 192-212 has been revised for clarity.

Small comments:

  1. 226 - Previous studies have suggested that vitamin E supplements needed to be 400 mg - This should be carefully discussed as doses ≥ 400 IU (400 mg of d/l tocopherol) can be associated with harmful effect - https://pubmed.ncbi.nlm.nih.gov/15537682/, discuss please.

Authors’ response: Lines 234-253 have been added to clarify that supplements are needed in these patients with fat malabsorption. Also please see our latest review: Traber, M.G.; Head, B. Vitamin E: How much is enough, too much and why! Free Radic Biol Med 2021, 177, 212-225, doi:10.1016/j.freeradbiomed.2021.10.028.

Remove sentence „Our previous work also demonstrated that cigarette smokers exhibited an accelerated turnover of tocopherols in the plasma compartment and that vitamin C supplements restored the turnover rate in the plasma to that of non-smokers [34,35,37].“ from rows 238-240. This is repetition of the introduction and here it does not bring a clear contribution to the discussion.

Authors’ response: The sentence has been deleted.

Just a curiosity, the clinical study was performed 2008-2009, what is the reason for such a long distance between the study and this paper?

Authors’ response: The data was calculated promptly, but authors were distracted by other duties and the manuscript languished in the drawer. The opportunity of publishing in an issue dedicated to cystic fibrosis prompted us to complete the manuscript.